# Current Knowledge on the Use of Neuromonitoring in Thyroid Surgery

**DOI:** 10.3390/biomedicines12030675

**Published:** 2024-03-18

**Authors:** Beata Wojtczak, Karolina Sutkowska-Stępień, Mateusz Głód, Krzysztof Kaliszewski, Krzysztof Sutkowski, Marcin Barczyński

**Affiliations:** 1Department of General, Minimally Invasive and Endocrine Surgery, Wroclaw Medical University, Borowska Street 213, 50-556 Wroclaw, Poland; karolina.sutkowska@onet.pl (K.S.-S.); krzysztof.kaliszewski@umw.edu.pl (K.K.); krzysztof.sutkowski@umw.edu.pl (K.S.); 2Infermedica Sp. z o.o., 50-062 Wroclaw, Poland; mateusz.glod@me.com; 3Department of Endocrine Surgery, Jagiellonian University Medical College, 50 Mikolaja Kopernika Street, 31-501 Krakow, Poland; marbar@mp.pl

**Keywords:** recurrent laryngeal nerves, thyroid surgery, intraoperative nerve monitoring, vocal fold paresis

## Abstract

Thyroid surgery rates have tripled over the past three decades, making it one of the most frequently performed procedures within general surgery. Thyroid surgery is associated with the possibility of serious postoperative complications which have a significant impact on the patient’s quality of life. Recurrent laryngeal nerve (RLN) palsy and external branch of the superior laryngeal nerve (EBSLN) palsy are, next to hypoparathyroidism and postoperative bleeding, some of the most common complications. The introduction of neuromonitoring into thyroid surgery, which enabled both the confirmation of anatomical integrity and the assessment of laryngeal nerve function, was a milestone that began a new era in thyroid surgery. The International Neural Monitoring Study Group has produced a standardization of the technique of RLN and EBSLN monitoring during thyroid and parathyroid surgery, which in turn increased the prevalence of neural monitoring during thyroidectomy. The current status of IONM and the benefits of its use have been presented in this publication.

## 1. Introduction

Thyroid surgery rates have tripled over the past three decades, making it one of the most frequently performed procedures within general surgery. Moreover, the increase in the incidence of papillary cancer, especially low-risk cases, have contributed to an increase in the number of thyroid surgeries performed [1,2]. 

Surgical treatment of thyroid diseases is associated with the possibility of serious postoperative complications which have a significant impact on the patient’s quality of life. Recurrent laryngeal nerve (RLN) palsy and external branch of the superior laryngeal nerve (EBSLN) palsy are, next to hypoparathyroidism and postoperative bleeding, some of the most common complications. While unilateral RLN injury usually results in hoarseness and changes in vocal timbre, bilateral injury may result in a complete loss of voice, shortness of breath, stridor and life-threatening acute respiratory failure. This condition often requires a tracheostomy. According to Jeannon et al., about 1 in 10 patients experience temporary recurrent laryngeal nerve injury after thyroid surgery, with longer-lasting voice problems in up to 1 in 25 cases [3]. EBSLN injury is the most underestimated vocal fold paralysis in the statistics of thyroid surgery complications. The frequency of this injury occurring during the procedure is estimated at 0.3–58% and its detection is practically impossible during postoperative laryngoscopy; therefore, complications related to this nerve are often overlooked. Its unilateral injury leads to changes in voice timbre and significantly weakens its strength. In turn, bilateral injury may cause a hoarse, monotonous voice which quickly weakens during vocalization. This injury is particularly burdensome for people working with their voice [4]. The frequency of vocal cord paralysis depends on the type of surgery (primary/secondary surgery), thyroid pathology (non-cancerous goiter/thyroid cancer/Graves’ disease), the extent of the surgery (partial/total removal of the thyroid gland) and the experience of the surgeon him or herself [3].

For years, the gold standard for preventing laryngeal nerve injury during thyroid surgery has been intraoperative identification of the laryngeal nerves. Visual identification of the RLN and the EBSLN allows only for the confirmation of the anatomical integrity of the laryngeal nerves, without the possibility of assessing their function. The introduction of neuromonitoring into thyroid surgery, which enabled both the confirmation of anatomical integrity and the assessment of laryngeal nerve function, was a milestone that began a new era in thyroid surgery, which has become much safer and more precise over the last three decades. The use of laryngeal nerve monitoring provides a chance to avoid the most serious complication in thyroid surgery, which is bilateral vocal fold paresis, by enabling the operator to finish the surgery after unilateral nerve injury detection (staged thyroidectomy) [4,5,6,7,8].

For over three decades, intraoperative RLN neuromonitoring as a standardized method has been increasingly used around the world. The benefits of using neuromonitoring in terms of its clinical, educational and legal aspects have determined the widespread acceptance of this method by both experienced surgeons and doctors in specialization training [5,6,7,8,9,10,11].

## 2. Anatomy of the Laryngeal Nerves

The surgical treatment of thyroid diseases has always been associated with the risk of voice disorders, which affect the proper functioning of a person in society, and for many, also their professional life. Voice disorders after thyroid surgery are the result of unintentional injury to the RLN or the EBSLN, which are located in the immediate vicinity of the thyroid gland [4].

The RLN branches from the vagus nerve at the level of the arch of the aorta on the left and the right subclavian artery on the right. The right nerve crosses the undersurface of the right subclavian artery and ascends in the neck to extend to the right tracheoesophageal groove. Usually, the nerve crosses superficially or deeply to the inferior thyroid artery or between its branches (Figure 1). The left RLN runs around the arch of the aorta and ascends more vertically in the left tracheoesophageal groove. The mean diameter of the RLN is about 1–3 mm; the left nerve is 12 cm in length and is longer than the right nerve, which is usually about 7 cm long. The nerve carries motor, sensory and parasympathetic fibers. There are many anatomic relationships and variations which make the nerve prone to injury during thyroidectomy. Anatomical variations of the RLN include the following: different course of the RLN at the level of the inferior thyroid artery hilum, the ligament of Berry, different course along tracheoesophageal groove, Zuckerkandle’s tubercle and a non-recurrent laryngeal nerve. Moreover, RLN branching is observed in 90% of cases and is another risk factor for RLN injury. Most often, the RLN divides into an anterior and posterior branch. The anterior branch supplies motor fibers to intrinsic laryngeal muscles. The posterior branch usually supplies sensory fibers to the trachea, esophagus and hypopharynx. A non-recurrent RLN is an extremely rare anatomical variant which occurs in 0.5–1% of cases on the right side and in less than 0.04% of cases on the left side. A non-recurrent RLN runs directly from the vagus nerve (usually at the level of the inferior thyroid artery) to the larynx. A left non-recurrent RLN is associated with dextrocardia and situs inversus (Figure 2). Detailed knowledge of the RLN’s anatomy is crucial for the surgeon to locate the nerve. Usually, the inferior thyroid artery, the trachoesophageal groove and the laryngeal entry point are landmarks for RLN identification [12,13].

The EBSLN is very close to the superior thyroid pedicle and poses a particular risk of injury during dissection of the vessels. The superior laryngeal nerve (SLN) is one of the first branches of the vagus nerve. The SLN divides into an internal and external branch, which descend dorsally to the carotid sheath and then extend to the larynx (Figure 1). After the EBSLN travels down the lateral surface of the larynx on the inferior pharyngeal constrictor muscle, the EBSLN typically bifurcates into two branches at the level of the cricoid cartilage, entering separately at the pars recta and pars obliqua of the cricothyroid muscle bellies. The sternothyroid–laryngeal triangle, named Joll’s space, is a landmark in EBSLN localization (Figure 3). The EBSLN is usually dorsal to the superior thyroid artery and superficial to the inferior pharyngeal constrictor muscle. The EBSLN is 0.8 mm wide and 8–8.9 cm long [6,12].

## 3. Mechanisms and Risk Factors of Nerve Injury

The most common mechanisms of recurrent laryngeal nerve injury include traction (e.g., during the extraction of the retrosternal goiter); incision or electrocoagulation are much less frequent mechanisms. Jeannon et al. showed, based on a review of 27 studies analyzing over 25,000 thyroid surgeries, that the average frequency of RLN paralysis was 9.8%. The complication rate ranged from 2.3–26% [3]. This large discrepancy is due to many factors; secondary operations on the thyroid gland, large goiter, retrosternal goiter, as well as operations for thyroid cancer and Graves’ disease have a higher rate of complications than procedures for multinodular goiters [14]. Moreover, the rate of RLN injury depends on the number of laryngeal examinations after thyroidectomy, which are not routinely performed in all surgery units.

Injury to the EBSLN can occur in up to 58% of patients [15,16,17].

The Cernea classification is the most common and widely recognized anatomical classification of the EBSLN based on the potential risk of injury to the nerve during thyroidectomy. This classification is as follows: Type 1: the EBSLN crosses the superior thyroid artery more than 1 cm above the upper edge of the thyroid superior pole and is common in 68% patients with small goiters and in 23% of patients with large goiters. Type 2A: the nerve crosses the superior vessels less than 1 cm above the upper edge of the superior pole; recognized in 18% of patients with small goiters and 15% of patients with large goiters. Type 2B: the EBSLN crosses the superior thyroid vessels below the upper edge of the superior thyroid pole. Usually, type 2B occurs in 14% of patients with small goiters and 54% with large goiters. Type 2B is the most prone to injury (Figure 2). Visual identification of the EBSLN may be impossible in up to 20% of patients when the nerve is located deep within the fascia of the inferior constrictor muscle [18].

## 4. Consequences and Diagnosis

Unilateral RLN palsy most often results in hoarseness or in voice timbre or swallowing disorders, while bilateral injury may result in shortness of breath and acute respiratory failure, which may be life-threatening. This condition often requires a tracheostomy. In more than 20% of patients, vocal cord paralysis resulting from RLN injury may be asymptomatic; therefore, laryngoscopy examination is the only objective tool for the proper assessment of the percentage of RLN palsy and should always be performed both before and after thyroid surgery [4,5].

In the case of EBSLN injury—the most underestimated complication in thyroid surgery—the patient is unable to produce high-pitched sounds and the voice weakens during modulation, which is important for people working with their voice [4,6]. Clinically, patients with EBSLN injury present with a hoarse or weak voice. These symptoms are the results of the dysfunction of the cricothyroid muscle, which is innervated by the EBSLN [6]. Injury to this nerve is difficult to identify in a routine postoperative laryngoscopy and both the Voice Handicap Index (VHI) and the Voice-Related Quality of Life instrument (V-RQOL) are validated instruments to assess the quality of voice and the risk of EBSLN injury [4].

Already in 1938, one of the pioneers of thyroid surgery, Frank Lahey, observed on the basis of over 3000 performed thyroidectomies that routine identification of the RLN during thyroid surgery reduces the frequency of its injury [19]. Currently, visual identification of the RLN is the gold standard in thyroid surgery, and for over 30 years, this method has been complemented by the use of intraoperative neuromonitoring of the RLN and the EBSLN during the procedure. The advantage of neuromonitoring of the laryngeal nerves over visual assessment alone is the ability to assess not only the preserved anatomical integrity of the nerve, but also its function during surgery [5,6,7,8].

## 5. Monitoring of the Laryngeal Nerves

### 5.1. History

Intraoperative neuromonitoring involves an electromyographic response from the vocal muscles after the electrical stimulation of the laryngeal nerves (RLN and EBSLN), which motorically innervate the vocal folds. Monitoring of the laryngeal nerves during thyroid surgery was first used by Shedd in 1966 [20]. Their paper published in “Annals of Surgery” presented the RLN and SLN monitoring in an animal model. In the same year, the author translated this study to the human population and showed how to monitor the RLN and SLN during thyroid surgery with the use of an endolaryngeal balloon [20]. However, it is most likely that Riddle was the first surgeon who identified the RLN using laryngeal palpation with stimulation of the RLN. In 1970, Riddell published a work based on 23 years of his own experience (from 1946 to 1986) [21]. Finally, in 1986, Galivan and Galivan showed that palpation of the posterior cricoarytenoid muscle combined with nerve stimulation of 0.5–2.0 mA is a safe technique for RLN identification and assessment during thyroid surgery [22]. Over the following years, various IONM techniques were used, including laryngeal palpation, glottic pressure monitoring, glottic observation and intralaryngeal hookwire electrodes [23]. Both the variety of techniques used and the lack of standardization meant that this technique did not come into common use until the beginning of the 21st century. There was a need for the standardization of IONM techniques to ensure that the results generated were repeatable, reliable and clinically meaningful [23].

In 2006, the International Neural Monitoring Study Group (INMSG) was established to serve the emerging field of neurophysiologic monitoring of the laryngeal nerves in neck endocrine surgery. It brought together experts to collaborate on improving the quality of IONM. The result of their cooperation was the standardization of RLN and EBSLN monitoring techniques during thyroid and parathyroid surgery [5,6]. In 2018, the INMSG published a two-part consensus guideline regarding staged bilateral thyroid surgery in the case of loss of signal (Part I) and optimal RLN monitoring for invasive thyroid cancer (Part II) [7,8]. Since then, the era of IONM in thyroid surgery has begun, starting its widespread use around the world.

### 5.2. Technique

Today, the most common system for intraoperative neuromonitoring is based on the use of an endotracheal tube with built-in surface electrodes. When the patient is intubated, the tube is precisely placed between the vocal folds (Figure 4). During thyroid surgery, using an electrical probe (mono- or bipolar), the nerve is stimulated with a current of 0.5–2.0 mA. After the latency period, a contraction of the vocal muscles occurs, which is detected by surface electrodes placed on the endotracheal tube and transmitted to the receiving part—the neuromonitor, which reflects a contraction in the form of an electromyographic (EMG) wave. Amplitudes of the EMG wave above 200 µV indicate the correct functioning of the nerve, and its absence at a current intensity of 1–2 mA indicates the so-called loss of signal (LOS) and laryngeal nerve injury. Detailed knowledge of the LOS resolution algorithm is essential for the surgeon to properly predict postoperative nerve function. This algorithm was discussed in detail in the recommendations regarding the use of neuromonitoring in thyroid surgery [5,6].

Cooperation with an anesthesiologist plays an important role in performing thyroid procedures using neuromonitoring of the laryngeal nerves. It is important to correctly place the appropriately selected endotracheal tube so that the surface electrodes adhere to the vocal folds. To obtain a response from the vocal muscles during laryngeal nerve stimulation, it is necessary to use short-acting muscle relaxants during intubation [5].

During nerve stimulation, modern devices generate both an acoustic signal and an electromyographic signal (EMG wave) on the monitor. The EMG wave recording proving the response from the laryngeal nerves should be archived and attached as a printout to the medical documentation.

During neuromonitoring-assisted surgery, a mapping technique is often used, which involves identifying the nerve in the surgical field by moving the stimulation probe at small, regular intervals (1–2 mm) along the trachea [5]. The signal of the evoked potential in the area of the nerve guides the operator to its location and allows for the proper determination of its course. The mapping technique is particularly applicable during secondary procedures on the thyroid gland where scar tissues have formed [24].

Monitoring of the laryngeal nerves is a standardized technique. This means that in all centers where this method is used, there is an adopted scheme of thyroid surgery using neuromonitoring, which includes laryngeal examinations performed before (L1) and after (L2) surgery, as well as identification and assessment of vagus nerve (V1) and laryngeal nerve (R1) activity both before and after thyroid lobe removal (V2 and R2, respectively) [5]. A laryngeal examination performed before surgical treatment makes it possible to diagnose even inconspicuous disorders of their functioning, which may often be asymptomatic; Moreover, correct phonation does not always indicate the absence of disorders in the functioning of the vocal folds. Therefore, a laryngeal examination performed after thyroid surgery is more important, as approximately 30% of vocal fold paralysis may occur with properly maintained phonation. Stimulation of the vagus nerve before identifying the RLN is performed to verify the correct positioning of the endotracheal tube with built-in surface electrodes, which determines the optimal use of the neuromonitoring technique. Stimulation of the vagus nerve after resection of the thyroid gland is the most sensitive way to assess the function of the RLN and excludes the possibility of its potential injury throughout its entire course (from its branching from the vagus nerve to its entry into the larynx) [4,5].

### 5.3. Types of IONM

At the moment, we have different options for RLN monitoring: intermittent intraoperative neuromonitoring (I-IONM) using a handled probe and continuous intraoperative neuromonitoring (C-IONM) using a temporary implantable vagus electrode [5,14,23]. The most recently introduced technique is Time Trend Monitoring, which seems to be a transitional technique between I-IONM and C-IONM [25].

Conventional intermittent IONM, the most common technique, can only provide intermittent RLN evaluation, allowing the nerve to be at risk of injury between the stimulations. C-IONM overcomes this limitation of I-IONM by offering real-time RLN monitoring using temporary vagus electrode stimuli. This technique involves placing a special receiving electrode on the vagus nerve at the beginning of the procedure, which allows for the continuous monitoring of RLN function. The advantage of this method over intermittent stimulation is that continuous stimulation can detect impending injury to the RLN, e.g., during goiter extraction from behind the sternum (enabling the surgeon to avoid excessive traction) [26,27]. The use of C-IONM allows the surgeon to perform corrective actions like stopping or reversing underlying maneuvers; thus, permanent injury may be avoided [14,23,26,27,28,29,30]. Phelan et al. (2014), considering adverse EMG events, has coupled amplitude decline and latency incline to identify mild combined events (mCEs) and severe combined events (sCEs) [29]. An mCE is defined as an amplitude decrease of 50–70% with a latency increase of 5–10%, and an sCE as an amplitude decrease of >70% and a latency increase of >10% (Figure 5).

He discovered that postoperative vocal fold palsy was not connected with mCEs, whereas sCEs could result in LOS and postoperative vocal fold paresis [29]. In this way, C-IONM is not only able to reduce bilateral vocal fold paresis, like I-IONM, but is also able to prevent unilateral permanent traction-related nerve injury. The natural evolution from I-IONM to C-IONM is Nerve Trend^TM^. In 2020, NIM Vital equipment (Medtronic, Jacksonville, FL, USA) was introduced to the market, offering NerveTrend^TM^ EMG reporting, which enables nerve condition tracking throughout a procedure, even when using I-IONM. It is operator-dependent and not automatic like C-IONM. This new IONM technique is used in the same manner as in the I-IONM arm and EMG trending includes amplitude and latency changes from initial vagal baseline of the NIM. The NerveTrend^TM^ mode operates at 3–5 min intervals or in cases of difficult maneuvers to assure an almost real-time EMG tracing and therefore allows for a tailored surgical approach by the modification of surgical maneuvers in case of occurrence of sCEs (yellow zone) in order not to end up with LOS (red zone), as shown in Figure 6 [25].

### 5.4. Use of IONM in EBSLN Assessment

Approximately 20% of EBSLNs cannot be identified by visualization alone because the nerve has a subfascial or intramuscular course within the inferior constrictor muscle. Nerve stimulation can objectively identify the EBSLN, leading to a visible cricothyroid muscle twitch in all (100%) cases. The application of IONM should not be limited to the RLN only but should be also expanded to EBSLN mapping, identification and functional testing during thyroidectomy [6].

In 2013, Barczyński published guidelines for IONM recording of EBSLN during thyroid and parathyroid surgery [6]. Before these recommendations, there had been few reports in the literature describing how to identify and preserve the EBSLN during superior pole dissection, and the identification of the EBSLN was not of interest to many surgeons [18,31,32]. There are two ways to monitor the EBSLN: (I) evaluation of the cricothyroid twitch response, which is present in all patients, and (II) EMG glottic response of vocal cord depolarization identified by surface ET electrodes. The latter maneuver is present in about 70–100% of patients depending on the type of ET used. Transverse division of the superior edge of the sternothyroid muscle and gentle traction of the superior thyroid pole into a lateral and caudal direction, followed by blunt dissection within the avascular plane of the sternothyroid–laryngeal triangle, allows for improving the exposure of the EBSLN, which usually descends parallel to the superior thyroid artery and lies on the fascia or between the fibers of the inferior constrictor muscle before its termination within the cricothyroid muscle.

## 6. Clinical Aspects of the Use of Neuromonitoring in Thyroid Surgery

The IONM should be utilized during thyroid and parathyroid surgery for many reasons, the most important of which are discussed below.

### 6.1. Complete Removal of Thyroid Tissue

In thyroid surgery, the basis for a properly performed operation is the complete removal of the pathological tissue (thyroidectomy is performed when changes in the thyroid gland affect both lobes, and lobectomy in case of changes located in one lobe only). Subtotal thyroid resection should not be performed due to the need for radicalization in the event of a postoperative diagnosis of thyroid cancer and due to the possibility of goiter recurrence. Secondary procedures on the thyroid gland have a much higher rate of complications, especially injury to the laryngeal nerves [24,33]. Neuromonitoring enables the precise dissection of the laryngeal nerves from the surrounding tissues [34] and significantly increases the radicalness of the surgeries performed. Barczyński et al. showed that the average iodine 131I uptake after a total thyroidectomy using intraoperative neuromonitoring compared to procedures during which no neuromonitoring was used was 0.67% ± 0.39% vs. 1.59% ± 0.69% (*p* < 0.001), and the percentage of patients with iodine uptake below 1% increased by as much as 45% when neuromonitoring was used [35]. The use of a careful dissection technique in the area of Zuckerkandl’s tubercle and Berry’s ligament with the use of neuromonitoring undoubtedly contributed to obtaining such good results [13,35].

Neuromonitoring allows us to predict postoperative RLN activity as early as during the operation, which practically eliminates the risk of bilateral damage [5,7,8,35,36]. The introduction of neuromonitoring to thyroid surgery established the concept of a two-stage thyroidectomy. It involves refraining from removing the second lobe of the thyroid gland if nerve injury is suspected on the side that is already being operated on (presence of signal loss on the originally operated side). This procedure is intended to protect the patient against bilateral RLN paresis and possible tracheostomy [35]. LOS troubleshooting algorithms have been developed to assist surgeons using IONM in identifying true LOS and determining the optimal course for any remaining parts of the surgical procedure. The introduction of C-IONM made it possible to avoid unilateral vocal fold paresis due to traction. Moreover, IONM is able to precisely identify the type of RLN injury; type I (focal nerve injury) and type II (global nerve injury) [5].

Additionally, an undeniable value of applying IONM is improving the rate of EBSLN identification.

### 6.2. IONM in the Reduction of RLN Injury

Since IONM was introduced into thyroid surgery, there has been an ongoing discussion about the superiority of IONM over visualization alone. Generally, the effectiveness of identifying the RLN using neuromonitoring is 98–100% and is statistically significantly higher compared to visual identification, which has been confirmed by numerous multicenter studies [37,38,39]. Moreover, in the vast majority of publications, lower rates of RLN injury are noted with IONM, but the differences are not statistically significant [40,41,42,43]. Dralle et al. pointed out that to reach an adequately powered study would require 9 million patients per arm for benign goiter and 40 thousand patients per arm for thyroid malignancy surgery to detect a statistically significant difference [40]. Moreover, another limitation of many publications is the heterogeneity of the studies, including variability in laryngeal examination practice, type and extent of the surgical procedure and surgeon’s experience [4,14,23]. The use of neuromonitoring reduces the number of postoperative vocal cord paralysis, which was first confirmed by Barczyński et al. in a randomized clinical trial involving the assessment of 2000 laryngeal nerves at risk of injury during 1000 thyroid surgeries [41]. The authors showed that the number of transient nerve palsies in people operated on with the use of neuromonitoring was 2.9% and 0.9% lower in the high-risk group (surgery for thyroid cancer with central lymphadenectomy, large, retrosternal goiter) and low-risk patients (multinodular goiter), respectively, compared to the number of these complications in patients undergoing a procedure during which only visual identification was used. In another large study analyzing over 850 cases of secondary thyroid operation, Barczynski et al. found a statistically significant reduction (2.6% vs. 2.4%) in transient paralysis rates between patients operated with vs. without I-IONM [42]. One of the first meta-analyses by Pisanu et al. from 2014 shoved no statistically significant difference in the incidence of RLN palsy when using IONM versus visualization alone during thyroidectomy, although in the IONM group, the rates of transient and permanent paresis were lower compared to visualization alone: 2.62% vs. 2.72% and 0.79% vs. 0.92%, respectively. This retrospective, observational study, which included 23,512 patients, must be approached with caution, as it was based mostly on non-randomized studies [43]. In 2017, Brandon published a high-quality PRISMA-compliant systematic review of overlapping meta-analyses summarizing the current state of I-IONM for the prevention of RLN injury during thyroidectomy and found that to date, I-IONM had not achieved a significant level of RLN injury reduction [44]. A meta-analysis by Yang et al. from 2017 showed reduced RLNP rates when using IONM, but without statistical significance for persistent RLN palsy [45]. An interesting meta-analysis on intraoperative neuromonitoring in high-risk thyroidectomy was published in 2017 by Wong et al. The authors compare the use of IONM with visual RLN identification alone during high-risk thyroidectomies: reoperations, thyroid surgery for malignancy, thyrotoxicosis and retrosternal goiter. The use of IONM decreased the rate of overall RLN palsy during reoperation (7.6% vs. 4.5%) and the result was statistically significant [46]. In 2023, Wojtczak et al. proved the superiority of IONM over visualization alone during thyroidectomy by analyzing risk factors in thyroid surgery. The authors proved that risk factors for complications in thyroid surgery are not significant for any increase in the rate of vocal fold paralysis as long as the surgery is performed with IONM, in contrast to thyroid surgery performed only with visualization alone [47]. In the context of reducing RLN injury, it is worth noting that various nerve monitoring techniques influence the rate of complications. In 2021, Schneider et al. proved the superiority of continuous over intermittent intraoperative nerve monitoring in preventing vocal cord palsy. Based on numbers of the nerves at risk (5208 vs. 5024 nerves), continuous IONM had a 1.7-fold lower early postoperative vocal cord palsy rate than intermittent monitoring (1.5 vs. 2.5%). This translated into a 30-fold lower permanent vocal cord palsy rate (0.02 vs. 0.6%). In multivariable logistic regression analysis, continuous IONM independently reduced early postoperative vocal cord palsy 1.8-fold (odds ratio (OR) 0.56) and permanent vocal cord palsy 29.4-fold (OR 0.034) compared with intermittent IONM. With continuous IONM, 1 permanent vocal cord palsy per 75.0 early vocal cord palsies was observed, compared with 1 per 4.2 after intermittent IONM. Therefore, early postoperative vocal cord palsies were 17.9-fold less likely to become permanent with continuous than intermittent IONM [48]. In 2023, the first randomized control trial by Barczyński and Konturek was published, comparing Nerve Trend vs. conventional I-IONM in the prevention of recurrent laryngeal nerve events during bilateral surgery. The study included 264 patients: 132 operated with Nerve Trend^TM^ vs. 132 operated with conventional I-IONM. On the first postoperative day, RLN injury was less frequent in the Nerve Trend^TM^ group (1.89% nerves at risk) vs. the I-IONM group (4.65%), but the results were not statistically significant (*p* = 0.67). Moreover, the study showed that staged thyroidectomy was not performed in the Nerve Trend^TM^ group (0%) vs. 4.54% of patients in the conventional I-IONM group and the results were statistically significant (*p* = 0.029). This study confirms the hypothesis that the use of the NerveTrend^TM^ mode results in reduced RLN injury on postoperative day 1 and significantly decreases the need for a staged thyroidectomy [25].

It is much easier to prove the superiority of IONM over visualization alone in preventing EBSLN injury. There are many publications proving that the use of IONM significantly improved the identification rate of the EBSLN during thyroidectomy, as well as reduced the risk of early phonation changes after thyroidectomy [41,49,50,51].

### 6.3. Loss of Signal during Thyroidectomy and Decision Making

The definition of loss of signal was proposed in the guidelines from 2011 [5]. Part I of the INMSG Guidelines discusses the use of IONM in bilateral thyroid surgery in the case of loss of signal and its incorporation into the surgical strategy. The INMSG recommends that neural monitoring information should be obtained and utilized in the strategy of a planned bilateral procedure by staging the surgery in the setting of ipsilateral LOS. This algorithm should be shared and discussed with the patient during the preoperative informed consent process. This is important because approximately 30% of patients with bilateral vocal fold palsy require tracheostomy. This strategy may prevent bilateral vocal fold palsy by allowing the surgery to be staged [7]. The second recommendation from Part I states that a surgeon should prioritize concern for the obvious significant medical and psychological morbidity of bilateral VCP and possible tracheotomy (even temporary) over perceived surgical convenience, the routine of performing the procedure “as planned” or the potential perceived impact on surgical reputation due to openly acknowledging the surgical complication of ipsilateral loss of signal. The full benefit of neural monitoring information in this surgical setting is appreciated through both the optimization of the patient’s quality of life as well as of surgical cost [7,52]. Part I of the Guidelines was published in conjunction with Part II of the INMSG Guidelines regarding the optimal management of a recurrent laryngeal nerve that is adherent to or invaded by cancer using preoperative glottic function through preoperative laryngeal examination as well as intraoperative monitoring electromyography signal [7,8]. To sum up, in the case of loss of signal without electromyography recovery, the surgeon should consider postponing the contralateral procedure to limit the risk of bilateral cord paralysis and tracheostomy.

### 6.4. Educational Value of Intraoperative Neuromonitoring: Research Studies

It should be noted that the identification of laryngeal nerves is difficult, especially for a young surgeon, as this skill is acquired over many years [53]. Neuromonitoring is of particular value to both young surgeons undergoing training and experienced surgeons who perform relatively few thyroid surgeries. An efficient performance of thyroid procedures by a surgeon requires more than 100 thyroidectomies per year [54]. Neuromonitoring enables young surgeons to learn about the anatomical variants of both nerves and their anatomical variability resulting from various factors (displacement of the trachea, retrosternal goiter, infiltration of thyroid cancer into the surrounding tissues). Moreover, one should not forget about possible anomalies, such as the identification of a non-recurrent laryngeal nerve—this is a very rare developmental anomaly detected on the right side in approximately 0.5% of patients [34].

Understanding the mechanisms of RLN injury during procedures influences the development of surgical techniques for a safe thyroidectomy, both for younger surgeons and for those with extensive experience [5].

With the introduction of neuromonitoring to thyroid surgery, numerous studies began in the field of the neurophysiology and neuropathology of the laryngeal nerves, which resulted in deepening our knowledge in these fields. Serpell et al. observed that in the case of RLN branching in the extralaryngeal section, RLN motor fibers responsible for the adduction and abduction of the vocal folds are most often located in the anterior branch of this nerve [36]. However, the proper use of IONM requires appropriate training. In 2021, the INMSG published a document which describes the minimum training required for learning the practical application of IONM. The training includes both basic and advanced courses ran by a specialized accreditation center [55].

## 7. Medical and Legal Aspects of Neuromonitoring Use in Thyroid Surgery

The use of intraoperative neuromonitoring provides a possibility to obtain medical documentation confirming the proper function of the RLN after the completed surgical procedure. This documentation may be archived as evidence of the utmost care in identifying and maintaining RLN activities during the operation.

Moreover, knowledge about the functioning of the vocal folds in the immediate postoperative period facilitates and directs the conversation with the patient in the event of hoarseness in the first days after the procedure. It allows for the differentiation of voice disorders dependent on RLN injury from abnormalities that may be caused by other factors (difficult intubation, postoperative swelling of the glottis, iatrogenic injury to the larynx, etc.).

The implementation of the neuromonitoring method in thyroid surgery has undoubtedly influenced the development of standards of conducting this type of procedure. It is also an important element of intraoperative surgical treatment quality control [5,6,56]. In 2021, the INMSG published the recommended informed consent for intraoperative neural monitoring in thyroid and parathyroid surgery. The INMSG consensus statement outlines the general and specific considerations regarding the surgical use of IONM and provides the essential recommended standard elements of informed consent for the use of IONM, thereby assisting surgeons and patients in the informed consent process and in shared decision making prior to thyroid or parathyroid surgery [57].

In Germany, intraoperative neuromonitoring is used over 100% of thyroid surgeries. The German Association of Endocrine Surgery Guidelines recommend the use of IONM in all cases of thyroid and parathyroid surgery [41,42,58]. Moreover, in Germany, there is the highest number of thyroid operations performed with C-IONM, reaching about 17% [11]. Other American and international guidelines recommending the use of IONM point to its utility in neural identification, reduction in transient nerve paralysis, prognostication of nerve function and avoidance of bilateral cord paralysis [9,59,60,61,62,63,64].

## 8. Prospects for the Development of Neuromonitoring around the World

Over the past three decades, we have observed an increasing interest in IONM all over the world. It is undoubtedly the fastest-developing technique in head and neck surgery [56,57,65]. According to the EUROCRINE register, IONM was used in 85.2% of thyroid surgeries in Europe in 2022, but most of the operations were performed with I-IONM (87.1%), whereas C-IONM was used in only 12.9% of thyroid operations [65].

A publication by Feng A.L. et al. from 2019 revealed an increased prevalence of neural monitoring during thyroidectomy according to a global surgical survey carried out among 1.015 respondents. The survey showed that 83% of respondents used IONM: 65.1% of them used it always and 18.1% selectively. In centers where IONM was used selectively, the main indications were reoperative (secondary) thyroid surgeries (95.1%) and cases of preoperative vocal cord paralysis (59.8%). Another conclusion from the study was that surgeons ≤ 45 years of age and those with ≤15 years of practice used IONM more than their peers (*p* < 0.001). Thyroid surgery volume, fellowship training and type of practice had no bearing on IONM use [10].

In Poland, the country where the authors work, over 30,000 thyroid surgeries are performed annually. It is the fourth most frequently performed general surgery procedure. Among over 300 surgical departments offering the treatment of thyroid diseases in 2011, only a few centers had the equipment for intraoperative RLN identification; therefore, only 1–2% of thyroid surgeries in Poland were neuromonitoring-assisted. Centers with neuromonitoring equipment used it primarily in patients with an increased risk of postoperative complications. After the first decade of neuromonitoring use in Poland, this method has become extremely popular—currently, 50% of thyroid surgeries are neuromonitoring-assisted. The use of neuromonitoring is not reimbursed by the National Health Fund, but this situation may soon improve due to the active work of the Polish Research Group for Neuromonitoring, established in 2010, which operates within the Polish Club of Endocrine Surgery. Its members have stated that endocrine surgery centers should be provided with neuromonitoring equipment. They also expressed a need to conduct routine training programs in the field of standardized laryngeal nerve neuromonitoring techniques and the use of this method during surgery for selected thyroid diseases.

## 9. Safety of Intraoperative Neuromonitoring

According to a publication by Caruso et al., it must be stated that intraoperative neuromonitoring of the superior and inferior laryngeal nerves, even with vagal stimulation, can be considered a safe method despite the complications related to it [66]. These complications are rare and if they do occur, they are usually related to displaced electrodes, the obstruction of the endotracheal tube or the drugs used for anesthesia. When IONM was first introduced, the effects of the stimulation on the nervous structures were considered. In 2012, Friedrich et al. established the safety of vagal stimulation during neuromonitoring in a non-randomized prospective study [67]. The authors analyzed changes in heart rhythm and immunomodulatory effects induced by vagal stimulation. Finally, no hemodynamic abnormalities were observed and no decrease in plasma levels of cytokines or TNF-α was observed after vagal stimulation. The standards for IONM, including gentle vagus and RLN stimulation, guarantee safety, and even older patients with advanced atrioventricular block could be safely monitored during continuous intraoperative nerve monitoring [68,69].

## 10. Conclusions

Neuromonitoring of the laryngeal nerves, both the recurrent nerves and the external branch of the superior laryngeal nerve, has become the standard in thyroid surgery. The use of the basic intermittent neuromonitoring technique (i-IONM) facilitates not only the identification of the nerve but also enables the assessment of its functional integrity, and not only its anatomical integrity, as in the case of the visual nerve identification method. In the event of the loss of neuromonitoring signal, it is possible to modify the surgical procedure, i.e., postpone the operation on the opposite side if the loss of signal on the first operated side persists during the procedure, which is essential in the prevention of bilateral RLN damage (so-called staged thyroidectomy). Continuous neuromonitoring (c-IONM) enables the surgeon to recognize impending RLN injury during surgery (most commonly caused by the traction mechanism), employ correct surgical maneuvers to avoid nerve damage and verify the recovery of RLN function after intraoperative electromyographic signal loss. Hence, the c-IONM technique has the potential to prevent not only bilateral but also unilateral injury to the recurrent laryngeal nerve. In centers that do not have the ability to use c-IONM, the NerveTrend mode may be helpful in reducing the risk of unilateral nerve damage and limiting the indications for staged thyroid surgery. In turn, EBSLN neuromonitoring is important for improving the quality of life of patients undergoing thyroid surgery, increasing the chance of maintaining their vocal timbre and register after surgery.

## Figures and Tables

**Figure 1 biomedicines-12-00675-f001:**
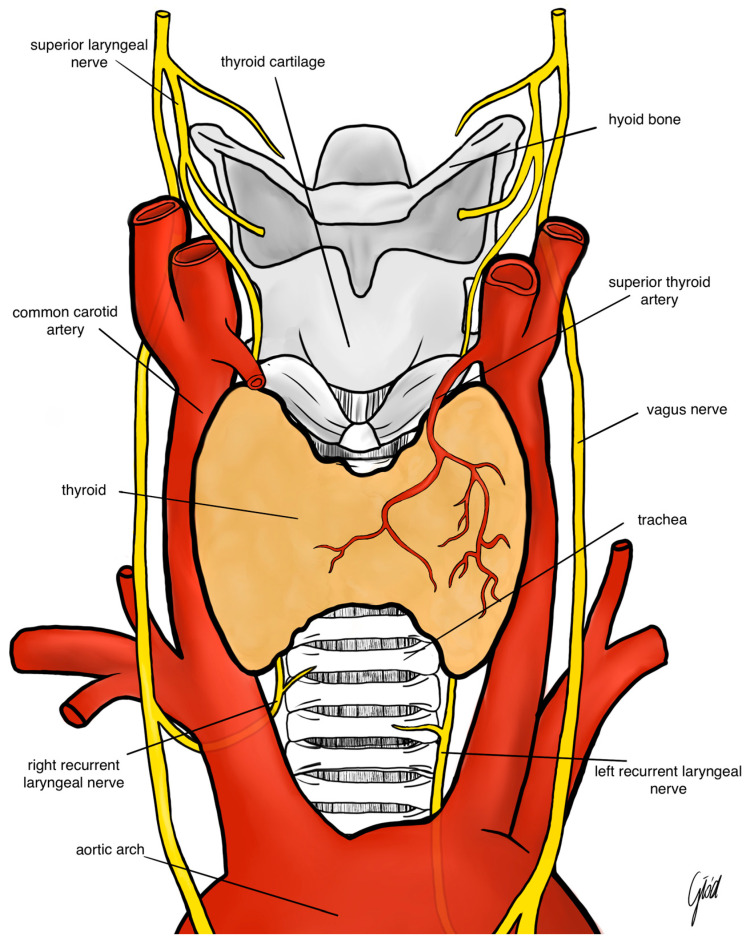
Anatomy of the recurrent laryngeal nerve and the superior laryngeal nerve.

**Figure 2 biomedicines-12-00675-f002:**
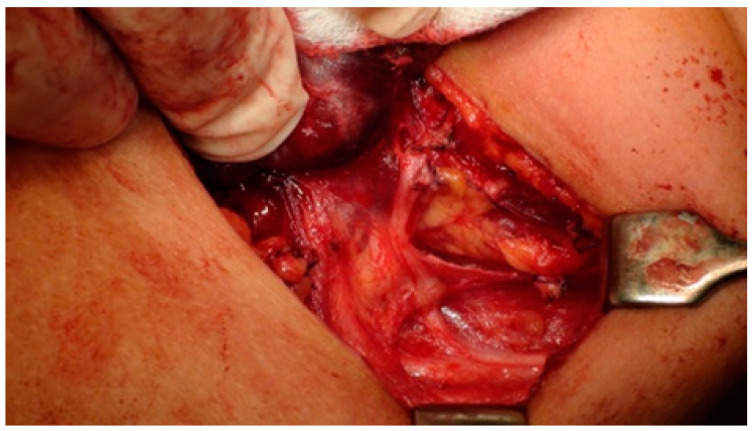
Non-recurrent laryngeal nerve.

**Figure 3 biomedicines-12-00675-f003:**
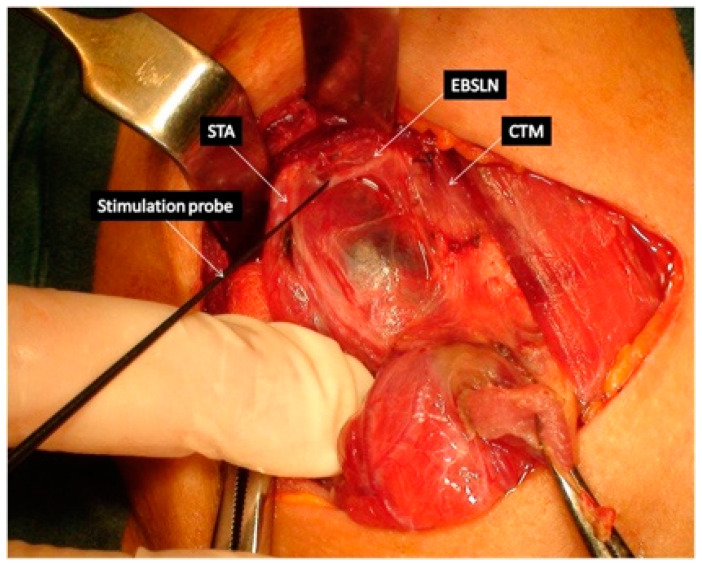
External branch of the superior laryngeal nerve. (STA—superior thyroid artery, EBSLN—external branch of the superior laryngeal nerve, CTM—cricothyroid muscle).

**Figure 4 biomedicines-12-00675-f004:**
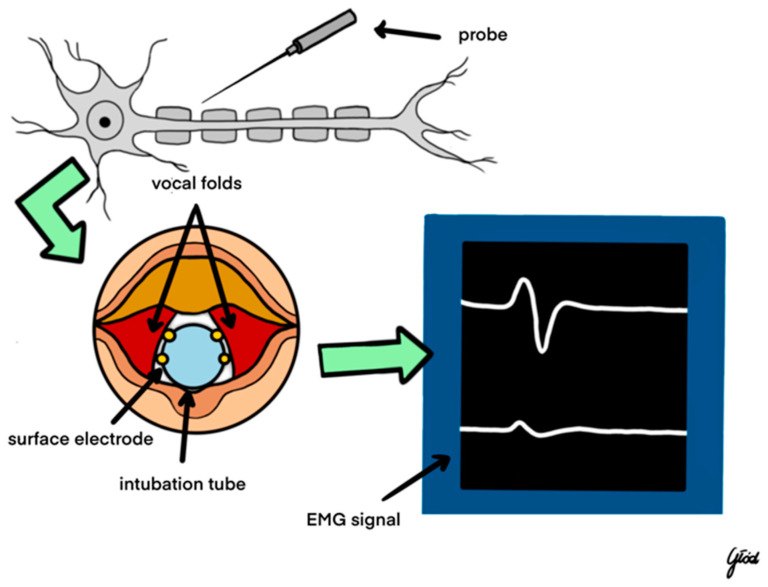
IONM—intraoperative nerve monitoring.

**Figure 5 biomedicines-12-00675-f005:**
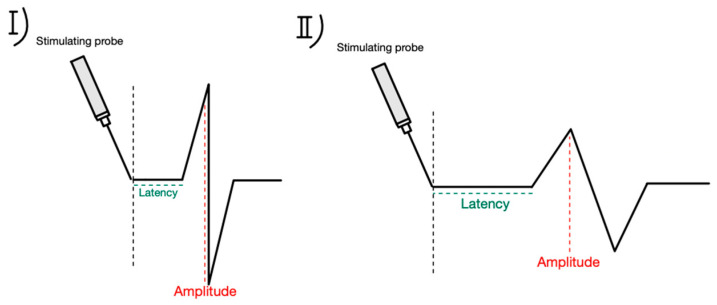
Amplitude and latency pattern during IONM. (**I**)—baseline latency and amplitude, (**II**)—amplitude—decreased, latency—increased.

**Figure 6 biomedicines-12-00675-f006:**
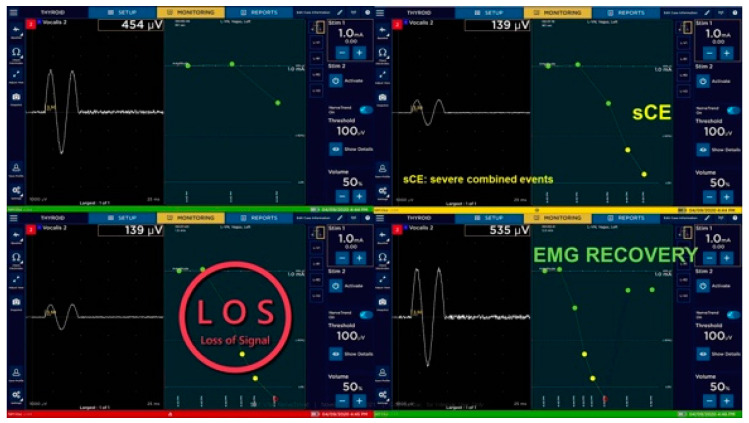
NIM NerveTrendTM mode: sCE followed by LOS and recovery of EMG signal can be recognized by the system to allow for intraoperative tailoring of surgical approach [courtesy of Marcin Barczyński].

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
