# Peer review of "Current Knowledge on the Use of Neuromonitoring in Thyroid Surgery"

_biomedicines, 2024, doi:10.3390/biomedicines12030675_

Round 1
Reviewer 1 Report
Comments and Suggestions for Authors
Your paper is a good chapter . Please just change"The Carnea classification " The Cernea Classification, You have ligthed up alçl the points in the ionm.
Author Response
Review 1:
Your paper is a good chapter . Please just change"The Carnea classification " The Cernea Classification, You have ligthed up alçl the points in the ionm.
Dear Reviewer,
I would like to thank you very much for taking the time to read the manuscript and for your valuable comments, of course I have changed “Carnea” into “Cernea” ( currently in Paragraph 157).
Moreover, english has been reviewed and moderate correction had been made.
Kind regards,
Beata Wojtczak
Reviewer 2 Report
Comments and Suggestions for Authors
THis is a nice comprehensive paper abput the IONM. The topic is actual. I have one recommendation: It would be more understandable to show some sample in Figure about the amplitude decrease and lateny increase.
Can you detail a little more the risk of the use of the IONM. Demage of the nerve?
Comments on the Quality of English LanguageEnglish is good, nicely understandable.
Author Response
There is the letter to the reviewer in an attatchment.

Reviewer 3 Report
Comments and Suggestions for Authors
I would like to thank the Editor for letting me review this interesting paper by Wojtczak et al.
This work provides up-to-date information and adequate references.
My concerns relate to the design of the manuscript.
Introduction is too long and repetitions between Introduction and other paragraph should be avoided.
The second paragraph called 'Thyroid surgery' is not adequate. It would be preferable to split in paragraphs like 'Anatomy of laryngeal nerves', 'Mechanisms and risk factors of nerve injury', 'Consequences and diagnosis'.
The 'Technique' paragraph could also be split into paragraphs regarding each technique or at least each nerve.
'Clinical aspects' shares information on the surgical technique (which could be part of the 'Technique' paragraph), and the roles in IONM (which could be part of the next paragraph).
Indeed, it is interesting to discuss the role of 'IONM in the reduction of RLN injury' in the fifth, but this could be associated in a paragraph presenting all aspects of the impact of IONM : complete removal of thyroid tissue, reduction of nerve injury, management of loss of signal, educational value...
'Medical and legal aspects' could include the guidelines presented in the penultimate paragraph.
Thus, I would recommand working on the desing of the manuscript in order to facilitate readers' comprehension and retention of important messages.
Figures are too small in the version of the manuscript I received and Figure 4 is reversed.
Comments on the Quality of English Language
English needs moderate editing.
Author Response
There is the letter to the reviewer in an attatchment
